# Autologous Human Dendritic Cells from XDR-TB Patients Polarize a Th1 Response Which Is Bactericidal to *Mycobacterium tuberculosis*

**DOI:** 10.3390/microorganisms13020345

**Published:** 2025-02-05

**Authors:** Rolanda Londt, Lynn Semple, Aliasgar Esmail, Anil Pooran, Richard Meldau, Malika Davids, Keertan Dheda, Michele Tomasicchio

**Affiliations:** 1Centre for Lung Infection and Immunity, Division of Pulmonology, Department of Medicine, University of Cape Town and UCT Lung Institute, Cape Town 7925, South Africa; 2South Africa MRC Centre for the Study of Antimicrobial Resistance, University of Cape Town, Cape Town 7925, South Africa; 3Institute of Infectious Diseases and Molecular Medicine, University of Cape Town, Cape Town 7925, South Africa; 4Faculty of Infectious and Tropical Diseases, Department of Immunology and Infection, London School of Hygiene and Tropical Medicine, London WC1E 7HT, UK

**Keywords:** dendritic cell vaccine, pre-extensively drug-resistant tuberculosis, extensively drug-resistant tuberculosis, drug-resistant tuberculosis, cellular vaccine, monocytic-derived dendritic cells

## Abstract

Extensively drug-resistant tuberculosis (XDR-TB) is a public health concern as drug resistance is outpacing the drug development pipeline. Alternative immunotherapeutic approaches are needed. Peripheral blood mononuclear cells (PBMCs) were isolated from pre-XDR/XDR-TB (*n* = 25) patients and LTBI (*n* = 18) participants. Thereafter, monocytic-derived dendritic cells (mo-DCs) were co-cultured with *M. tb* antigens, with/without a maturation cocktail (interferon-γ, interferon-α, CD40L, IL-1β, and TLR3 and TLR7/8 agonists). Two peptide pools were evaluated: (i) an ECAT peptide pool (ESAT6, CFP10, Ag85B, and TB10.4 peptides) and (ii) a PE/PPE peptide pool. Sonicated lysate of the *M. tb* HN878 strain served as a control. Mo-DCs were assessed for DC maturation markers, Th1 cytokines, and the ability of the DC-primed PBMCs to restrict the growth of *M. tb*-infected monocyte-derived macrophages. In pre-XDR/XDR-TB, mo-DCs matured with *M. tb* antigens (ECAT or PE/PPE peptide pool, or HN878 lysate) + cocktail, compared to mo-DCs matured with *M. tb* antigens only, showed higher upregulation of co-stimulatory molecules and IL-12p70 (*p* < 0.001 for both comparisons). The matured mo-DCs had enhanced antigen-specific CD8^+^ T-cell responses to ESAT-6 (*p* = 0.05) and Ag85B (*p* = 0.03). Containment was higher with mo-DCs matured with the PE/PPE peptide pool cocktail versus mo-DCs matured with the PE/PPE peptide pool (*p* = 0.0002). Mo-DCs matured with the PE/PPE peptide pool + cocktail achieved better containment than the ECAT peptide pool + cocktail [50%, (IQR:39–75) versus 46%, (IQR:15–62); *p* = 0.02]. In patients with pre-XDR/XDR-TB, an effector response primed by mo-DCs matured with an ECAT or PE/PPE peptide pool + cocktail was capable of restricting the growth of *M. tb* in vitro.

## 1. Introduction

Tuberculosis (TB), caused by infection from the bacillus *Mycobacterium tuberculosis* (*M. tb*), is one of the leading causes of death from a single infectious agent worldwide and remains the deadliest infectious disease globally. In 2023, an estimated 1.09 million deaths among HIV-negative and 161,000 deaths among HIV-infected people were reported. South Africa is among the 14 countries with the highest burden of TB, TB/HIV, and drug-resistant (DR) TB [1].

Globally and in South Africa, the increasing prevalence of extensively drug-resistant TB (XDR-TB) is posing a significant threat to the control of TB [1]. XDR-TB is defined as resistance to (i) rifampicin and isoniazid from the first line treatment regime, (ii) any fluoroquinolone, and (iii) at least one of the drugs bedaquiline and linezolid. Globally, 159,684 cases of multi-drug-resistant TB (MDR-TB; resistance to rifampicin and isoniazid), and 28,982 cases of pre-XDR-TB (resistance to rifampicin and isoniazid and at least one of the fluoroquinolones) or XDR-TB were detected during 2022 [1]. The typical treatment duration for XDR-TB, even with newer drugs, is between 18 and 24 months. It is associated with a high pill burden, accompanied by frequent drug toxicity, high rates of adverse events, and low treatment success rates [2]. Treatment success in South Africa is 53% for XDR-TB [3]. Many patients who do not succumb to the disease, fail treatment and are discharged back into their communities [4,5], posing a significant risk for onward transmission [4]. Alternate treatment options are, therefore, urgently required because the drug development pipeline is slow moving, and as new drugs are developed, drug resistance amplification is rapidly evolving.

DC-based immunotherapy is a cellular therapeutic vaccination strategy whereby disease-specific ex vivo-generated autologous monocytic-derived dendritic cells (mo-DCs) are administered to the patient with the intention of eliciting protective and targeted cytotoxic T lymphocyte (CTL) immunity [6]. A DC-based immunotherapeutic intervention may offer potential benefits as an adjunctive treatment for XDR-TB. Immunotherapeutic strategies aim to improve overall immunity by reducing pathology associated with chronic disease, shorten treatment duration that will ultimately reduce treatment costs, and potentially increase treatment compliance leading to improved success rates [7,8,9]. Furthermore, immunotherapy offers a unique appeal for drug-resistant TB as new treatment regimens for these patients are limited [7,8,10,11].

In this study, mo-DCs were cultured from peripheral blood mononuclear cells (PBMCs) obtained from pre-XDR/XDR-TB patients and persons with latent TB infection (LTBI). Naturally occurring DCs account for less than 0.5% of the peripheral blood leukocytes; therefore, mo-DCs [12], which are functionally similar to naturally occurring inflammatory DCs, were used [13,14,15]. The mo-DCs were matured with *M. tb*-specific peptide pools, with or without a maturation cocktail [IFN-γ, IFN-α, CD40L, IL-1β, Ampligen (TLR-3 agonist), and R848 (a TLR7/8 agonist)] [12]. We show that we can optimally mature patient-specific mo-DCs in vitro with *M. tb*-specific peptide pools and a full maturation cocktail. The mature mo-DCs were able to prime *M. tb*-specific effector cells and polarise polyfunctional T-cell and CTL responses that were bactericidal to *M. tb* in vitro.

## 2. Materials and Methods

### 2.1. Study Site and Population

Pre-XDR-TB was defined as resistance to rifampicin and isoniazid, and at least one of the fluoroquinolones, while XDR-TB included additional resistances to one injectable [8]. Patients (*n* = 25) were recruited at Brooklyn Chest Hospital, Cape Town, South Africa, between 2017 and 2020 if they were >18 years old and healthy. HIV-infected persons with a CD4 count greater than 200 cells/mL were included. Latent TB-infected (LTBI) participants (*n* = 18) were recruited as controls if they were asymptomatic, had no previous history of TB, and were QuantiFERON-TB Gold In-Tube Interferon-Gamma Release Assay (QFT-GIT IGRA) positive. Please refer to the inclusion and exclusion criteria in the Appendix. Ethical approval was obtained from the Human Research Ethics Committee (HREC) of the University of Cape Town (#213/2016).

### 2.2. M. tb-Specific Antigens

The ECAT peptide pool comprised peptides from the ESAT-6, CFP-10, Ag85B, and TB10.4 proteins (Table A1). The PE/PPE peptide pool consisted of peptides from the PE_PGRS33, PE_PGRS62, PE18, PPE25, PPE33, and PPE46 proteins (Table A2). The ECAT [16,17,18,19,20,21] and PE/PPE peptides [22,23,24,25,26,27,28] were selected based on their immunogenicity supported in the literature. A sonicated lysate from a *M. tb* clinical Beijing stain, HN878, was the antigen positive control. The optimal concentration for both the ECAT and PE/PPE peptide pools were 1 µg/mL and 3 µg/mL for the HN878 lysate, determined using an IFN-γ enzyme-linked immune absorbent spot (ELISA; Mabtech, Cincinnati, OH, USA) assays.

### 2.3. Venipuncture and Peripheral Blood Mononuclear Cell Isolation

PBMCs were isolated via density centrifugation (using Leucosep^®^ tubes and Histopaque-1077 (Sigma-Aldrich, St. Louis, MO, USA) according to the manufacturers’ instructions (Greiner, Germany). The buffy coat, from centrifuged whole blood, was diluted with Dulbecco’s phosphate-buffered solution (DPBS, Lonza, Switzerland) and transferred to the Leucosep^®^ tube for centrifugation. Thereafter, the enriched cell fraction was harvested, washed with DPBS, and re-suspended. Cell density and viability were determined with Türks solution (Sigma-Aldrich, USA) or Trypan blue solution (Sigma-Aldrich, USA), respectively, under a light microscope using a haematocytometer.

### 2.4. Culture Conditions to Obtain Mature mo-DCs

Monocytes (~1 × 10^7^ cells) were isolated by plastic adherence and differentiated into immature mo-DCs with CellGenix DC medium (CellGenix, Freiburg, Germany) containing 100 µg/mL IL-4 and GM-CSF (Prospec Bio, East Brunswick, NJ, USA) for 5 days at 37 °C. After 5 days, the immature mo-DCs were incubated with or without 1 µg/mL ECAT or PE/PPE peptide pool, or 3 µg/mL HN878 lysate for 6 h at 37 °C and then matured with or without a full maturation cocktail containing (i) 25 ng/mL IFN-γ, (ii) 10 ng/mL IFN-α, (iii) 10 ng/mL IL1-β, (iv) 1 µg/mL CD40L (all purchased from Prospec Bio, East Brunswick, USA), (v) 100 µg/mL Ampligen (AIM ImmunoTech, Ocala, FL, USA ), and (vi) 2.5 µg/mL R848 (InvivoGen, San Diego, CA, USA) for 42 h at 37 °C. Untreated, immature DCs and DCs matured with a limited maturation cocktail, containing only 25 ng/mL IFN-γ, 10 ng/mL IFN-α, 10 ng/mL IL1-β, and 1 µg/mL CD40L, served as experimental controls. The TLRs were not included in the limited cocktail as TLR agonists are known to mature DCs [29,30]. The composition and concentration of the maturation cocktail was optimized in a previously published study conducted by Tomasicchio et al. [12]. Mo-DCs were stained for flow cytometry analysis and supernatants derived from the immature and mature mo-DCs were stored at −80 °C for IL-12p70 and IL-10 analysis by ELISA. A schematic of the experimental procedure used in this study can be found in Figure 1. The DC culture conditions are abbreviated in the text below as follows: iDCs = immature DCs; ECAT only = mo-DCs matured with ECAT peptide pool only; PE/PPE only = mo-DCs matured with PE/PPE peptide pool only; HN878 only = mo-DCs matured with sonicated HN878 lysate only; LC = mo-DCs matured with limited cocktail only; ECAT + C = mo-DCs matured with ECAT peptide pool and full maturation cocktail; PE/PPE + C = mo-DCs matured with PE/PPE peptide pool and full maturation cocktail; and HN878 + C = mo-DCs matured with sonicated HN878 lysate and full maturation cocktail.

### 2.5. Generation of Effector Cells

Mature mo-DCs were co-cultured with autologous PBMCs to generate DC-primed effector cells, as previously described [12] (Figure 1). Briefly, mature mo-DCs were co-cultured at a ratio of 1:10 in RPMI (Lonza, Basel, Switzerland) medium supplemented with 10% human A/B serum (Western Province Blood Transfusion Services, South Africa), 2 mM L-glutamine, 25 mM HEPES, 0.1 mg/mL sodium pyruvate, 100 IU/mL penicillin, and 100 mg/mL streptomycin (R-10; Sigma, Buchs, Germany). After 3 days, the medium was replaced with fresh medium containing 10 U/mL IL-2 (Roche, Basel, Switzerland). The cells were cultured for an additional 4 days at 37 °C to generate effector cells. DC-primed effector cells were stained for flow cytometry analysis and culture supernatants were stored at −80 °C for cytokine and chemokine analysis by Luminex assay.

### 2.6. IL-12p70 and IL-10 ELISA

The expression of IL-12p70 and IL-10 was determined from the culture supernatants using ELISA, according to the manufacturer’s specifications (Mabtech, Stockholm, Sweden).

### 2.7. Sample Preparation for Flow Cytometry

Cells were washed with DPBS then stained with viability stain for 15 min. Zombie Aqua Fixable viability kit (Biolegend, San Diego, CA, USA) was used for both DCs and the DC-primed effector cells. The immature and mature DCs were stained for HLA-DR-PerCP/Cyanine 5.5, CD40-FITC, CD80-PE/Cy7, CD83-APC, CD86-PE/Dazzle 594, CCR7-PE, PDL1-BV421, MMR-BV785, DC-SIGN-APC/FIRE 750, and TLR2-BV711 in BV Brilliant Stain buffer (Becton-Dickenson, Franklin Lakes, NJ, USA) for 30 min. Thereafter, they were washed and fixed with 1% Cell Fix solution (Becton-Dickenson, USA). The DC-primed effector cells were surface stained for CD3-AF700, CD4-APC/Cy7, CD8-PE/Cy5, CD45RO-BV650, CD69-PE/Cy7, and PD1-BV711 for 30 min. Thereafter, they were fixed in a 1% Cell Fix solution for 30 min, experienced permeabilization with a 1% Permeabilising solution 2 (Becton-Dickenson, USA) for 30 min, and were stained for Granulysin-PE, Perfoin-AF488, TNFα-BV785, IL17-PE/Dazzle 594, IL2-BV421, and IFN-γ-APC in BD Brilliant stain buffer for 45 min. The cells were washed and stored in a 4% paraformaldehyde solution in PBS (Becton-Dickenson, USA). All samples were acquired on a BD LSR-II within 3 days. The Minimum Information about a Flow Cytometry Experiment (MIFlowCyt) standards were followed to ensure consistency and reproducibility. Briefly, single-stained compensation controls, consisting of positive and negative beads, were used in this study. The compensation matrix was determined and applied using the FACS Diva software (version 7). All flow cytometry data were represented as % of a total population, normalizing the data for variability associated with absolute cell numbers. Flow cytometry data were analysed using FlowJo software (v9.9, FlowJo LLC, Ashland, OR, USA). The gating strategy for the immature and mature DCs as well as the DC-primed effector cells are presented in Figure A2 and Figure A3, respectively. All fluorescently labelled antibodies used are shown in Table A3 and Table A4.

### 2.8. Cytokine and Chemokine Luminex Assay

IL-10, TNF-α, IFN-γ, IL-13, RANTES, IL-17, and IL-6 were measured using a Luminex Milliplex kit (Merck, Darmstadt, Germany), according to the manufacturer’s instructions.

### 2.9. ESAT-6 and Ag85B Tetramer Assay

The MHC-1-specific tetramers were HLA-02 positive; therefore, only patient samples with matching HLA-02 type were included in the assay. Only DC-primed effector cells loaded with ECAT peptide pool and HN878 lysate were assessed for specificity. Effector cells were stained with Ag85B-PE tetramer, ESAT-6-APC tetramer (MBL, Ottawa, IL, USA), CD3-AF700, CD8-FITC (Becton Dickinson, Franklin Lakes, NJ, USA), and Zombie NIR (Biolegend, San Diego, CA, USA) as recommended by the manufacturer, followed by acquisition using a flow cytometry and analyses.

### 2.10. Mycobacterial Stasis Assay

A previously described mycobacterial containment assay was used to assess the mo-DC-primed effector cell responses on their ability to contain *M. tb* within peripheral blood monocyte-derived macrophages (MDMs) [31]. DC-primed effector cells were generated by co-culturing matured mo-DCs with autologous PBMCs for seven days (Figure 1). In parallel, 1 × 10^6^/mL PBMCs were seeded into a tissue culture plate and cultured for five days. Thereafter, the plate was washed to remove non-adherent cells and the MDMs were infected with H37*Rv* at an MOI of 5:1 for 18 h. The cells were further washed to remove any mycobacteria that was not taken up. On day seven, the DC-primed effector cells were co-cultured with *M. tb*-infected MDMs for 24 h. Colony-forming units (CFUs) were enumerated in all wells. The reference control contained *M. tb*-infected MDMs only and represented 100% *M. tb* survival. The percentage (%) mycobacterial containment was reported and defined as the reduction in *M. tb* survival compared to the reference control representing 100% *M. tb* survival.

### 2.11. Statistical Analysis

Data were analysed for statistical significance by one-way Analysis of Variance (ANOVA) with Dunnett’s post-test where applicable. A Wilcoxon signed-rank paired t-test was used to assess the differences within population groups. A Mann–Whitney U-test was used to assess the differences between population groups. Data were analysed using Graphpad Prism software (Version 10 GraphPad Software, La Jolla, CA, USA) where *, **, ***, and **** indicated *p* < 0.05, *p* < 0.01, *p* < 0.005, and *p* < 0.0001, respectively.

## 3. Results

### 3.1. Demographics and Clinical Characterisation of the Pre-XDR/XDR-TB Patients

The demographics of the pre-XDR (*n* = 4)/XDR-TB (*n* = 21) patients used for this study are shown in Table 1. The median age of the pre-XDR/XDR-TB patients was 34 years. Patients were more likely to be HIV negative and the majority (84%) of the patients had confirmed XDR-TB. Nine out of the 25 patients had a previous episode of TB. Three patients had previous drug-susceptible TB, five had MDR-TB, and one patient had a previous episode of XDR-TB. The median ranges of the patients’ haematological counts were within normal ranges. The patient and sample recruitment schematics for the pre-XDR-TB/XDR-TB patients and LTBI participants are shown in Figure 2 and Figure A1, respectively.

### 3.2. mo-DCs from Pre-XDR/XDR-TB Patients, Matured with M. tb Antigens and Full Maturation Cocktail, Expressed High Levels of Key Co-Stimulatory Molecules

Functional matured mo-DCs are characterized by the upregulation of C-C chemokine receptor 7 (CCR7), which is essential for migration, and CD80, CD83, and CD86 as key DC maturation markers (Figure 3A–D).

In LTBI participants and pre-XDR/XDR-TB patients, mo-DCs matured with ECAT peptide pool and cocktail (ECAT + C), PE/PPE peptide pool and cocktail (PE/PPE + C), or HN878 lysate and cocktail (HN878 + C) expressed higher levels of CCR7, CD80, CD83, and CD86 compared to mo-DCs matured with ECAT peptide pool only (ECAT only), PE/PPE peptide pool only (PE/PPE only), and HN878 lysate only (HN878 only; *p* ≤ 0.02 for all comparisons in LTBI; *p* ≤ 0.003 for all comparisons in pre-XDR/XDR; Figure 3A–D and Figure A4). Mo-DCs matured with the limited cocktail only (no antigen added) (LC), only upregulated CCR7, CD80, CD83, and CD86 to a similar degree to mo-DCs matured with ECAT + C, PE/PPE + C, or HN878 + C.

### 3.3. Matured mo-DCs from Pre-XDR/XDR-TB Patients Expressed High Levels of Th1 Polarising Cytokines

IL-12p70 is a Th1 polarizing cytokine that is necessary for the generation of a protective immune response [32].

In LTBI participants and pre-XDR/XDR-TB patients, mo-DCs matured with ECAT + C, PE/PPE + C, or HN878 + C expressed higher levels of IL-12p70 compared to mo-DCs matured with the peptide pool (ECAT or PE/PPE) or lysate only (*p* ≤ 0.03 for all comparisons in LTBI; *p* = 0.004 for all comparison in pre-XDR/XDR-TB; Figure 4A and Figure A5).

IL-10 is a characteristic Th2 polarizing cytokine. In pre-XDR/XDR-TB patients, mo-DCs matured with ECAT + C, PE/PPE + C, or HN878 + C produced higher levels of IL-10 compared to mo-DCs matured with ECAT only (*p* = 0.008), PE/PPE only (*p* = 0.004), or HN878 only (Figure 4B; *p* = 0.004). In LTBI participants, only mo-DCs matured with ECAT + C and HN878 + C produced higher levels of IL-10 compared to the peptide pool and lysate-only controls (*p* = 0.03 for all comparisons). In both LTBI participants and pre-XDR/XDR-TB patients, there was no difference in the levels of IL-12p70 and IL-10 from mo-DCs matured with the LC to mo-DCs matured with ECAT + C, PE/PPE + C, or HN878 + C.

In order to determine the Th-polarizing response, the IL-12p70 (Th1)/IL-10 (Th2) ratio was determined (Figure 4C). Only DCs from pre-XDR/XDR-TB patients, matured with ECAT + C, PE/PPE + C, or HN878 + C, polarize a dominant Th1 response compared to DCs matured with the peptide pool and lysate-only controls (Figure 4C; *p* ≤ 0.03 for all comparisons).

### 3.4. Matured mo-DCs from Pre-XDR/XDR-TB Patients Primed CD4^+^ T-Cells That Expressed High Levels of Th1 and Polyfunctional Cytokines

The mo-DCs were co-cultured with autologous PBMCs to generate DC-primed CD4^+^ and CD8^+^ T-cells. Thereafter, the CD4^+^ T-cells were examined for single and polyfunctional expression of IFN-γ, TNF-α, IL-2, and IL-17 (Figure 5).

In pre-XDR/XDR-TB patients, CD4^+^ T-cells primed by mo-DCs matured with ECAT + C expressed higher levels of IFN-γ and IL-2 compared to CD4^+^ T-cells primed by mo-DCs matured with ECAT only (IFN-γ: *p* = 0.04; IL-2: *p* = 0.05). CD4^+^ T-cells primed by mo-DCs matured with HN878 + C expressed higher levels of IFN-γ and TNF-α compared to CD4^+^ T-cells primed by untreated mo-DCs (IFN-γ: *p* = 0.04; TNF-α: *p* = 0.004). Similarly, CD4^+^ T-cells primed by mo-DCs matured with HN878 only, expressed higher levels of single secreting IL-2 and TNF-α compared to CD4^+^ T-cells primed by untreated mo-DCs (IL-2: *p* = 0.05; TNF-α: *p* = 0.008). CD4^+^ T-cells primed by mo-DCs matured with the LC only upregulated IFN-γ (*p* = 0.008) and IL-2 (*p* = 0.02) compared to CD4^+^ T-cells primed by untreated mo-DCs. Nevertheless, compared to CD4^+^ T-cells primed by mo-DCs matured with ECAT + C or HN878 + C, there was no statistical difference between the IFN-γ, IL-2, and TNF-α expression primed by CD4^+^ T-cells primed by mo-DCs matured with the LC only.

For polyfunctionality, CD4^+^ T-cells primed by mo-DCs matured with HN878 + C expressed higher levels of co-secreting IFN-γ and TNF-α, IL-2 and TNF-α, and IFN-γ, TNF-α, and IL-2 compared to CD4^+^ T-cells matured with untreated mo-DCs (Figure 5; *p* < 0.008 for all comparisons). Similarly, CD4^+^ T-cells primed by mo-DCs matured with HN878 only expressed higher levels of co-secreting IFN-γ and IL-2, IFN-γ and TNF-α, IL-2 and TNF-α, and IFN-γ, TNF-α, and IL-2 compared to CD4^+^ T-cells matured with untreated mo-DCs (*p* < 0.05 for all comparisons). Interestingly, in LTBI participants, CD4^+^ T-cells primed by antigen-matured mo-DCs did not display a significant increase in cytokine expression.

### 3.5. Matured mo-DCs from Pre-XDR/XDR-TB Patients Primed CD8^+^ T-Cells That Expressed High Levels of Cytolytic Markers

The DC-primed CD8^+^ T-cells were assessed for granulysin and perforin expression (Figure 6).

In pre-XDR/XDR-TB patients, CD8^+^ T-cells primed by mo-DCs matured with the LC only and ECAT + C expressed higher levels of perforin compared to CD8^+^ T-cells primed by untreated mo-DCs (*p* ≤ 0.009). Mo-DCs matured with HN878 + C, however, primed CD8^+^ T-cells that expressed cytolytic markers to a greater degree. CD8^+^ T-cells primed by mo-DCs matured with HN878 + C expressed higher levels of granulysin and perforin compared to CD8^+^ T-cells primed by untreated DC (*p* = 0.01 for both comparisons). In LTBI participants, CD8^+^ T-cells primed by antigen-matured mo-DCs did not significantly upregulate granulysin, but upregulation of perforin was observed for HN878 + C only (*p* = 0.05).

### 3.6. Matured mo-DCs from Pre-XDR/XDR-TB Patients Primed Effector Cells That Secreted High Levels of Soluble Cytokines

The DC-primed effector cells were assessed for their secretion of IL-13, TNF-α, IL-6, IL-10, IL-17, and chemokine regulated upon activation, normal T-cell expressed and secreted (RANTES) (Figure 7A–D).

In pre-XDR/XDR-TB patients, effector cells primed by mo-DCs matured with ECAT + C secreted higher levels of IL-6 and RANTES compared to effector cells primed by mo-DCs matured with ECAT only (IL-6: *p* = 0.008; RANTES: *p* = 0.05). Effector cells primed by mo-DCs matured with PE/PPE + C secreted higher levels of IL-6 and IL-13 compared to effector cells primed by mo-DCs matured with PE/PPE only (IL-6: *p* = 0.02; IL-13: *p* = 0.03). Effector cells primed by mo-DCs matured with HN878 + C secreted higher levels of TNF-α, IL-6, and RANTES compared to effector cells primed by untreated mo-DCs (TNF-α: *p* = 0.02; IL-6: *p* = 0.01; RANTES: *p* = 0.003). The effector cells primed by mo-DCs matured with HN878 lysate only were not tested for soluble cytokine secretion. The effector cells primed by mo-DCs from LTBI participants were not tested for soluble cytokine secretion.

IL-13, TNF-α, IL-6, and RANTES secretion by the effector cells primed by mo-DCs matured with ECAT + C, PE/PPE + C, or HN878 + C was compared. Mo-DCs matured with ECAT + C primed effector cells secreted higher levels of TNF-α compared to effector cells primed by mo-DCs matured with PE/PPE + C (IL-13: *p* = 0.02; TNF-α: *p* = 0.2; IL-6: *p* = 0.02).

### 3.7. Matured mo-DCs from Pre-XDR/XDR-TB Patients Primed Antigen-Specific CD8^+^ T-Cells

Next, we wanted to determine if the mo-DCs have the ability to present *M. tb* antigens, and prime T-cell receptors (TCRs) of CD8^+^ T-cells using a tetramer assay. The ESAT-6 tetramer detected the TCR on CD8^+^ T-cells primed by mo-DCs matured with HN878 lysate only (*p* = 0.02) or HN878 + C (*p* = 0.008) compared to unstimulated PBMCs (Figure 8). ESAT-6 recognition was superior to Ag85B recognition. The Ag85B tetramer detected the TCRs on CD8^+^ T-cells primed by mo-DCs matured with HN878 only (*p* = 0.02), or HN878 + C (*p* = 0.008) compared to unstimulated PBMCs. Additionally, compared to unprimed CD8^+^ T-cells, Ag85B tetramer significantly detected the TCR on CD8^+^ T-cells primed by mo-DCs matured with HN878 only (*p* = 0.018) or HN878 + C (*p* = 0.008). CD8^+^ T-cells primed by mo-DCs from LTBI participants were not tested for antigen specificity.

### 3.8. Matured mo-DCs from Pre-XDR/XDR-TB Patients Primed Effector Cells That Were Bactericidal to M. tb In Vitro

Using a mycobacterial containment assay, the DC-primed effector cell responses were assessed for their ability to prime a functional and protective effector response (Figure 9B).

In pre-XDR/XDR-TB patients, mo-DCs matured with *M. tb* antigens only (collective responses from the ECAT peptide pool, PE/PPE peptide pool, and HN878 lysate) were compared to mo-DCs matured with *M. tb* antigens + C (grouped) to determine the impact of the cocktail on the containment efficacy (Figure 9A). Only mo-DCs matured with *M. tb*-antigen + C were effective at priming an effector response that restricted *M. tb* growth in vitro (*p* = 0.0002). Lastly, the containment effects of the ECAT + C and PE/PPE + C were compared (Figure 9B). Effector cells primed by mo-DCs matured with PE/PPE + C were more effective at containing the growth of *M. tb* in vitro compared to effector cells primed by mo-DCs matured with ECAT + C (*p* = 0.02).

In LTBI participants, the effector cells primed by mo-DCs matured with ECAT + C, PE/PPE + C, or HN878 + C did not significantly contain the growth of *M. tb* in vitro compared to mo-DCs matured with the peptide pool and lysate-only controls (Figure A6). However, when the containment effects of ECAT + C were compared to those of PE/PPE + C, the DC-primed effector responses from ECAT + C mediated a higher degree of containment than the DC-primed effector from PE/PPE + C (*p* = 0.02).

## 4. Discussion

We wanted to determine the feasibility of a DC immunotherapeutic intervention against *M. tb* in patients with pre-XDR/XDR-TB. Mo-DCs were successfully matured with either ECAT peptide pool, PE/PPE peptide pool, or sonicated HN878 lysate and the full maturation cocktail. The matured mo-DCs (i) upregulated key migratory and co-stimulatory molecules, (ii) produced high levels of the Th1 polarizing cytokine, IL-12p70, (iii) polarized a dominant Th1 response, (iv) primed CD4^+^ T-cells that produced high levels of expressed and soluble cytokines, (v) primed antigen-specific CD8^+^ T-cells that recognized ESAT-6 and Ag85B, and (vi) expressed high levels of cytolytic markers. The DC-primed effector cell response was shown to be functional, and, most importantly, had the ability to contain the growth of *M. tb*-infected macrophages in vitro.

We show that matured mo-DCs expressed a higher level of CCR7 compared to the peptide pool and lysate-only controls. CCR7 mediates the migration of DCs from the sites of antigen entry to the lymphoid organs for T-cell priming [33,34]. Its expression is a pre-requisite in DC immunotherapy [34,35] because high levels of CCR7 expression directly translate to survival benefits in cancer patients [36]. Effective T-cell priming at the lymph nodes requires the upregulation of co-stimulatory signalling and the secretion of Th1 polarizing cytokines [13,37,38]. The matured mo-DCs from pre-XDR/XDR-TB patients were shown to express high levels of CD80, CD83, and CD86 and secrete IL-12p70, a key Th1 polarizing cytokine. IL-12p70 secretion promotes sustained antigen-specific CTL activity of NK cell responses and positively correlates with favourable clinical outcomes such as overall survival in cancer patients [39,40,41]. The mo-DCs secreted only trace amounts of IL-10, a key Th2 polarizing cytokine; hence, a dominant Th1 response was shown. Importantly, the DC maturation results show that irrespective of the *M. tb* antigen/s used, it was the presence of the maturation cocktail that promoted the generation of phenotypically matured DCs.

Next, we characterized the DC-primed effector cells for cytokine secretion and cytolytic marker expression. CD4^+^ T-cells primed by matured mo-DCs expressed high levels of IFN-γ, IL-2, and TNF-α and primed high levels of polyfunctional T-cells, predominately IFN-γ, IL-2, and/or TNF-α. Several recent TB vaccine studies have utilized the induction of polyfunctional T-cells as a measure of a proxy for vaccine efficacy [16,42,43,44,45]. The MVA85A vaccine was a promising candidate for a novel TB vaccine as it induced potent polyfunctional CD4^+^ T-cell responses [21], yet it was unsuccessful in a phase 2a study [46]. Even though polyfunctional T-cells might not correlate with protective immunity, they are likely to be one part of a complex response that defines protective immunity against TB [47,48]. In our study, the DC-primed effector cells were shown to secrete high levels of soluble TNF-α, IL-6, and RANTES, highlighting that matured mo-DCs from pre-XDR/XDR-TB patients polarize a Th1 response. Even though IL-13 was detected, the cytokine secretion profile for the DC-primed effector cells skewed towards a dominant pro-inflammatory response. The matured mo-DCs from pre-XDR/XDR-TB patients had the ability to prime antigen-specific T-cell responses. The DC-primed CD8^+^ T-cells recognized ESAT-6- and Ag85B-specific tetramers, proving that patient-derived mo-DCs did present the selected antigens. Lastly, the DC-primed CD8^+^ T-cells expressed high levels of perforin and granulysin, which are essential for the effective neutralization of *M. tb* [49,50].

In pre-XDR/XDR-TB patients, mo-DCs matured with PE/PPE + C achieved a higher magnitude of containment compared to the mo-DCs matured with ECAT + C (50% versus 46%, *p* = 0.02). PE/PPE was, therefore, the most effective at promoting a functional immune response that restricted the growth of *M. tb* in vitro. This is in contrast to the data presented for the DC-primed effector cell responses, which suggested that the ECAT peptide pool was the more immunogenic peptide pool in terms of CD4^+^ cytokine expression and soluble secretion. Due to the nature of the ECAT and PE/PPE peptide pools, specifically that they consisted of 15mer or 9mer peptides, there is a binding affinity towards the MHC-II and MHC-I, respectively (Table A6). This could explain the contradictory results and provide a rationale for the slightly better containment efficacy of the PE/PPE peptide pool. The containment results further suggest that patient-specific cells from the pre-XDR/XDR-TB group may be contributing to their effector cell responses. In LTBI participants, the matured mo-DCs displayed variable and overall minimal efficacy at containing *M. tb* growth in vitro. Nevertheless, LTBI-derived mo-DCs, matured with ECAT + C, were able to mediate a higher degree of containment than mo-DCs matured with PE/PPE + C (*p* = 0.05). In both LTBI participants and XDR-TB patients, mo-DCs matured with *M. tb* antigens only were not successful in promoting effector responses that contained *M. tb* in vitro. It is, therefore, clear that *M. tb* antigens in conjunction with the full maturation cocktail are necessary for the maturation of mo-DCs that are able to promote a functional immune response.

The selection of the peptides used in this study was based on proven immunogenicity in the literature. ESAT-6, CFP-10, Ag85B, and TB10.4 used in the ECAT peptide pool are highly immunogenic and known to polarize a Th1 response [16,17,18]. These secreted peptides are associated with the ESX type VII secretion systems and have been incorporated in previous TB vaccine candidates [51]. ESAT-6 and CFP-10 are highly recognized by T- and B-cells in active TB patients and induce strong IFN-γ responses [44,52,53,54,55,56], while Ag85B is broadly recognized by CD4^+^ T-cell epitopes in active TB patients [20,44,56,57]. Similarly, TB10.4 elicits a strong response from both CD4^+^ and CD8^+^ T-cells in TB patients [18,57,58,59,60]. The PE/PPE peptide pool used in this study consisted of peptides from PE_PGRS33, PE_PGRS62, PE18, PPE25, PPE33, and PPE46. PE_PGRS33, PE_PGRS62, and PPE46 have been investigated as potential vaccine targets. They were shown to induce substantial antigen-specific proliferation and IFN-γ production [23,24,25] as well as induce strong humoral responses in patients with TB [61,62]. PE18 and PPE25, associated with ESX5, are potent immunodominant T-cell antigens [27] with high sequence homology to related proteins [27]. ESX-associated PE/PPE proteins exhibit cross-reactivity with numerous PE/PPE proteins and induce Th-1 polarizing cytokines in mice [26,27]. PPE33, a non-secreted protein, associated with ESX5, activates T-cells [63] and is expressed at higher levels in *M. tb* compared to *M. bovis* [64,65,66].

The mo-DCs were assessed for their involvement in the immune-suppressive PD-L1/PD-1 pathway (Figure A7 and Figure A8). Matured mo-DCs from LTBI participants showed no differences in PD-L1 compared to mo-DCs matured with *M. tb* antigens only (Figure A7). However, in pre-XDR/XDR patients, matured mo-DCs upregulated PD-L1 expression compared to mo-DCs matured with *M. tb* antigens only (*p* < 0.0001), but not significantly more when compared to LTBI. While PD-L1 expression is associated with immune-suppressive pathways, on DCs, it was shown to protect them from cytotoxic-mediated T-cells [67]. It is, therefore, possible that the upregulation of PD-L1 on matured mo-DCs serves a regulatory function. Furthermore, CD4^+^ T-cells primed by matured mo-DCs from pre-XDR/XDR-TB patients (*p* ≤ 0.05) and LTBI participants (*p* ≤ 0.05) upregulated PD-1 (Figure A8). The upregulation of PD-1 has been shown to occur on activated, antigen-presenting T-cells [68,69].

This study has several limitations. We did not show that PE/PPE-matured mo-DCs could present antigens using PE/PPE-specific tetramers, as these are not commercially available. The roles of regulatory T-cells (Tregs) upregulated in active TB patients [70] and other immunoregulatory pathways were not investigated. However, we did investigate PD1/PD-L1 regulatory pathways. Despite showing that PD1 and PD-L1 were upregulated on T-cells and DCs respectively, mo-DCs are associated with in vivo clinical efficacy [35,39], and our results show that XDR-TB patient-derived DCs can restrict mycobacterial growth in vitro. The containment assay could not distinguish between killed or non-replicating organisms or quantify anti-mycobacterial activity by cell type. However, containment is a much more biologically meaningful outcome measure than immunological biomarkers. The ECAT and PE/PPE peptide pools present with an affinity towards MHC-II and MHC-I, respectively. We do, however, show that both peptide pools achieved containment of *M. tb*-infected monocyte-derived macrophages, albeit better in the PE/PPE pool. HLA typing for patients was limited to resource constraints. This study included HIV co-infected patients (CD4 > 200 cells/mL), who constitute as much as 77% of cases in South Africa [71], to establish proof-of-concept for a cellular intervention in the vulnerable group. In general, the cytokine responses were lower in the HIV-infected versus uninfected persons, but this was only significant for IL-2. The levels of containment were slightly lower for ECAT + C in the HIV-infected versus HIV-uninfected persons, but this was not significant (Table A5).

## 5. Conclusions

In conclusion, we have cultured mo-DCs using PBMCs from patients with pre-XDR/XDR. Our most important finding was that the mo-DCs matured with an ECAT peptide pool, a PE/PPE peptide pool, or a HN878 lysate, and a full maturation cocktail was bactericidal to *M. tb* in vitro. The PE/PPE peptide pool was the most effective *M. tb*-specific peptide pool at containing *M. tb* in vitro. mo-DCs have been utilized as immunotherapeutic agents in a number of studies, albeit mostly cancer but, to our knowledge have not been investigated for their potential in patients with TB.

## Figures and Tables

**Figure 1 microorganisms-13-00345-f001:**
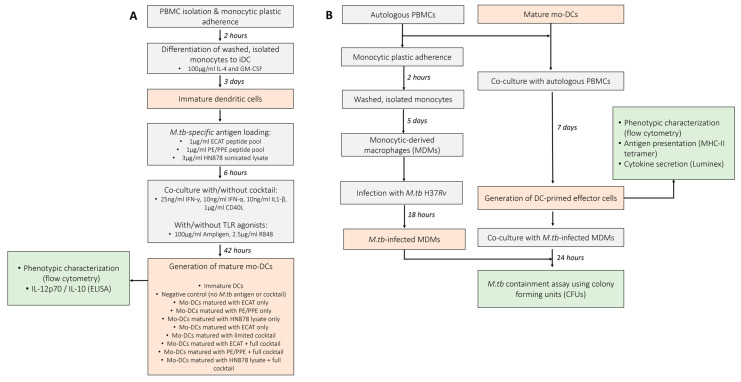
Schematic of the experimental procedures used to determine the efficacy of mo-DCs against *M. tb.* (**A**) Monocytes were differentiated into immature DCs with GM-CSF and IL-4 for five days. The immature DCs were loaded with *M. tb* antigens for six hours, then incubation with/without a maturation cocktail for two days. Immature and mature mo-DCs were stained for flow cytometry analysis and culture supernatants assessed for soluble cytokine secretion using ELISA. (**B**) For the generation of effector cells, PBMCs were co-cultured with mature mo-DCs for seven days. DC-primed effector cells were stained for flow cytometry, the culture supernatants assessed for soluble cytokine secretion using Luminex, and the DC-primed effector cells assessed for antigen specificity using a tetramer assay. After seven days, the mo-DC-primed effector T-cells were co-cultured with *M. tb* H37*Rv*-infected macrophages and *M. tb* containment (CFU/mL) determined using a stasis assay by counting colonies on agar plates. Abbreviations: ECAT = peptide pool consisting of ESAT-6, CFP-10, AG85b, and TB10.4 peptides; PE/PPE = peptide pool consisting of PE and PPE peptides; HN878 = sonicated lysate of *M. tb* HN878; LC = limited maturation cocktail containing IFN-γ, IFN-α, IL1-β, and CD40L; C = full maturation cocktail containing IFN-γ, IFN-α, IL1-β, CD40L, Ampligen (TLR3 agonist), and R848 (TLR7/8 agonist).

**Figure 2 microorganisms-13-00345-f002:**
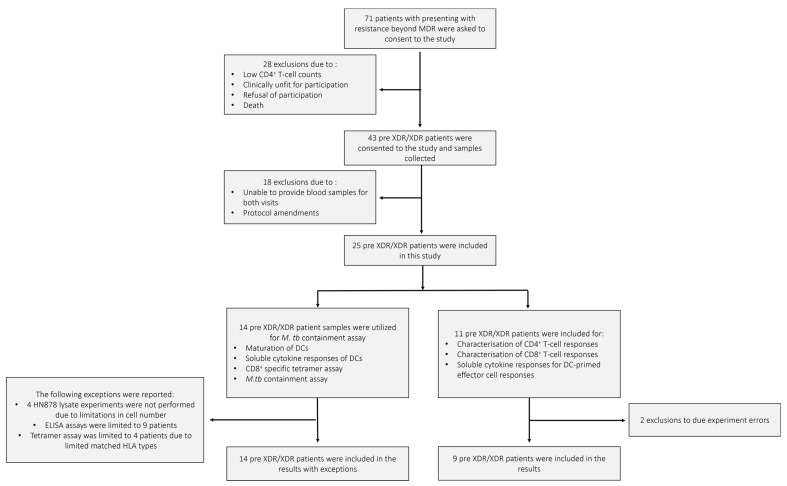
The pre-XDR/XDR-TB patient recruitment and sample schematic. A total of 71 individuals presenting with resistance beyond MDR were asked to consent to the study. After screening, only 43 were eligible for consent and sample collection. In total, 18 patients were further excluded due to failure to provide adequate blood volumes for the follow-up visit or amendments to the experimental protocol. A total of 25 pre-XDR/XDR-TB patient samples were included in the final results. Samples from 14 pre-XDR/XDR-TB patients were used for the phenotypic assessment of mature mo-DCs using flow cytometry, the determination of soluble cytokine secretion of mature mo-DCs using ELISA, the CD8^+^ specific tetramer assay for the determination of antigen specificity using flow cytometry, and the *M. tb* containment assay. Exceptions for the aforementioned experiments are reported. Samples from 11 pre-XDR/XDR-TB patients were used for the characterisation of DC-primed CD4^+^ and CD8^+^ T-cell responses and the determination of soluble cytokine secretion from the DC-primed effector cell responses. Exclusions due to experimental errors are reported.

**Figure 3 microorganisms-13-00345-f003:**
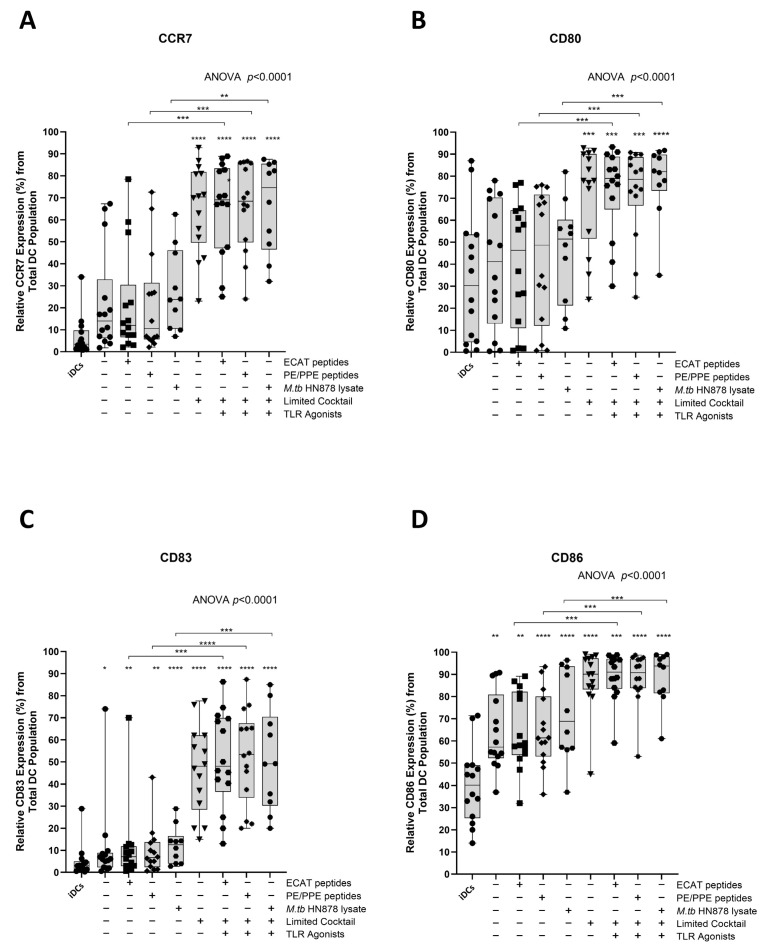
Matured mo-DCs from pre-XDR/XDR-TB patients expressed high levels of co-stimulatory molecules. Immature mo-DCs from pre-XDR/XDR-TB patients (*n* = 14) were differentiated from monocytes with IL-4 and GM-CSF for five days. After five days, the immature mo-DCs were matured with *M. tb* antigens, with/without the maturation cocktail for 48 h. The mo-DCs were analysed for CCR7 (**A**), CD80 (**B**), CD83 (**C**), and CD86 (**D**) by flow cytometry. The experimental conditions are immature DC, negative control (DCs matured without *M. tb* antigens or cocktail), mo-DCs matured with ECAT only, mo-DCs matured with PE/PPE only, mo-DCs matured with HN878 only, mo-DCs matured with limited cocktail only, mo-DCs matured with ECAT + C, mo-DCs matured with PE/PPE + C, and mo-DCs matured with HN878 + C. Statistical analysis was performed using GraphPad Prism. One-way ANOVA with Dunnett’s post-test was used to compare experimental groups to the control group (immature mo-DCs). For paired comparisons between mo-DCs matured with *M. tb* antigens only and mo-DCs with *M. tb* antigens + C, the Wilcoxon signed-rank test was applied. Significance levels are indicated as *, **, ***, **** for *p* < 0.05, *p* < 0.01, *p* < 0.005 and *p* < 0.0001 respectively. Outliers are represented by symbols, which indicate data points that fall significantly outside the interquartile range. Abbreviations: ECAT = peptide pool consisting of ESAT-6, CFP-10, AG85b, and TB10.4 peptides; PE/PPE = peptide pool consisting of PE and PPE peptides; HN878 = sonicated lysate of *M. tb* HN878; LC = limited maturation cocktail containing IFN-γ, IFN-α, IL1-β, and CD40L; C = full maturation cocktail containing IFN-γ, IFN-α, IL1-β, CD40L, Ampligen (TLR3 agonist), and R848 (TLR7/8 agonist).

**Figure 4 microorganisms-13-00345-f004:**
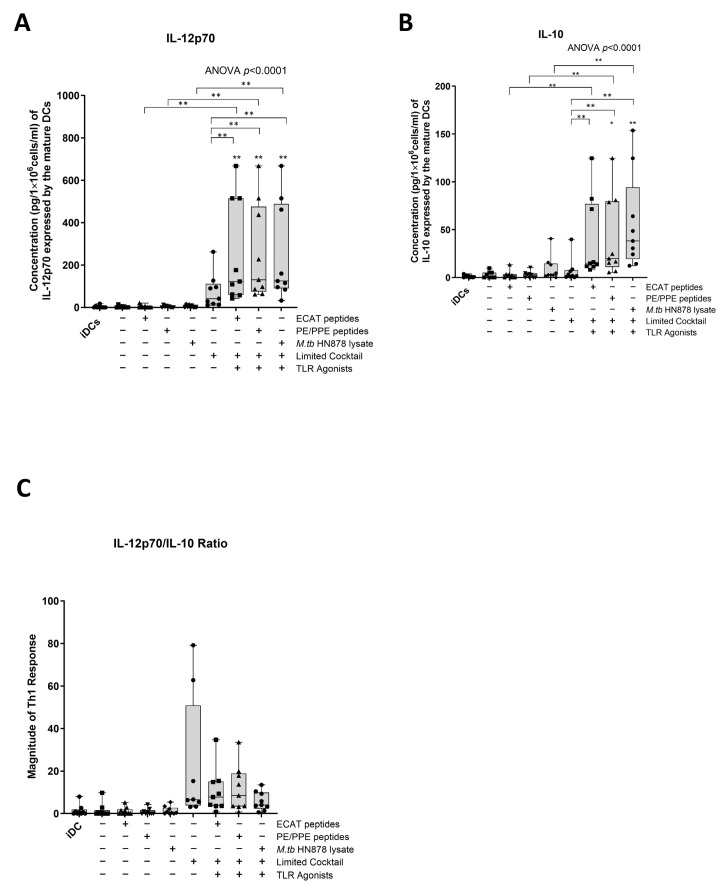
Matured mo-DCs from pre-XDR/XDR-TB patients polarized a Th1 response. The levels of IL-12p70 (**A**) and IL-10 (**B**) from the culture supernatants (*n* = 9) of the immature and matured mo-DCs were determined using the ELISAPRO IL-12p70 and IL-10 detection kit (Mabtech). In order to determine the Th-polarizing response, the IL-12p70/IL-10 (**C**) ratio was assessed. The experimental conditions are immature DC, negative control (DCs matured without *M. tb* antigens or cocktail), mo-DCs matured with ECAT only, mo-DCs matured with PE/PPE only, mo-DCs matured with HN878 only, mo-DCs matured with limited cocktail only, mo-DCs matured with ECAT + C, mo-DCs matured with PE/PPE + C, and mo-DCs matured with HN878 + C. Statistical analysis was performed using GraphPad Prism. One-way ANOVA with Dunnett’s post-test was used to compare experimental groups to the control group (immature mo-DCs). For paired comparisons between mo-DCs matured with *M. tb* antigens only and mo-DCs with *M. tb* antigens + C, the Wilcoxon signed-rank test was applied. Significance levels are indicated as * and ** for *p* < 0.05 and *p* < 0.01 respectively. Outliers are represented by symbols, which indicate data points that fall significantly outside the interquartile range. Abbreviations: ECAT = peptide pool consisting of ESAT-6, CFP-10, AG85b, and TB10.4 peptides; PE/PPE = peptide pool consisting of PE and PPE peptides; HN878 = sonicated lysate of *M. tb* HN878; LC = limited maturation cocktail containing IFN-γ, IFN-α, IL1-β, and CD40L; C = full maturation cocktail containing IFN-γ, IFN-α, IL1-β, CD40L, Ampligen (TLR3 agonist), and R848 (TLR7/8 agonist).

**Figure 5 microorganisms-13-00345-f005:**
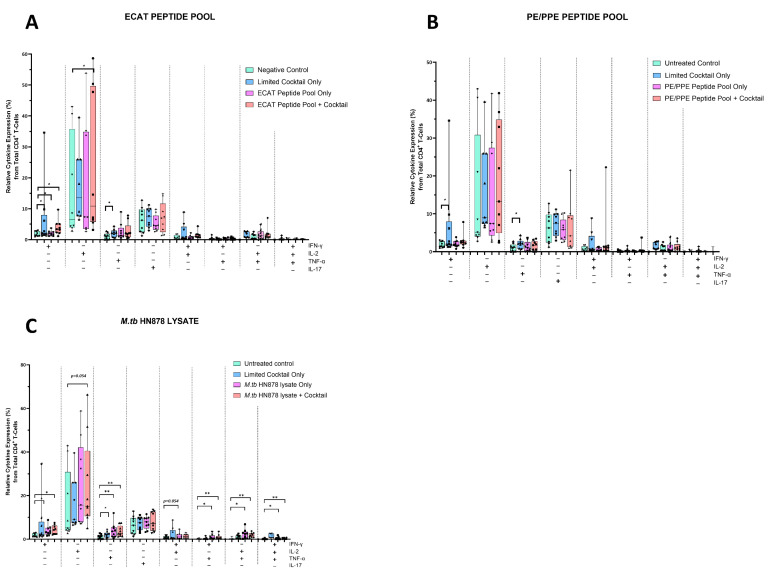
Matured mo-DCs from pre-XDR/XDR-TB patients primed CD4^+^ that expressed high levels of Th1 effector cytokines. PBMCs were co-cultured with XDR-TB patient-derived, matured mo-DCs (*n* = 9) for seven days. The single (IFN-γ, TNF-α, and IL-2) and polyfunctional (IFN-γ/IL-2; IFN-γ/TNF-α; IL-2 /TNF-α and IFN-γ/IL-2/TNF-α) cytokine secretion from the DC primed CD4^+^ T-cells, matured with either the ECAT peptide pool (**A**), the PE/PPE peptide pool (**B**) or the *M. tb* HN878 lysate (**C**) was determined by flow cytometry. The flow cytometry data were analysed using FlowJo. Statistical analysis was performed using GraphPad Prism. One-way ANOVA with Dunnett’s post-test was used to compare experimental groups to the control group (immature mo-DCs). For paired comparisons between mo-DCs matured with *M. tb* antigens only and mo-DCs with *M. tb* antigens + C, the Wilcoxon signed-rank test was applied. Significance levels are indicated as * and ** for *p* < 0.05 and *p* < 0.01 respectively. Outliers are represented by symbols, which indicate data points that fall significantly outside the interquartile range. Abbreviations: ECAT = peptide pool consisting of ESAT-6, CFP-10, AG85b, and TB10.4 peptides; PE/PPE = peptide pool consisting of PE and PPE peptides; HN878 = sonicated lysate of *M. tb* HN878; LC = limited maturation cocktail containing IFN-γ, IFN-α, IL1-β, and CD40L; C = full maturation cocktail containing IFN-γ, IFN-α, IL1-β, CD40L, Ampligen (TLR3 agonist), and R848 (TLR7/8 agonist).

**Figure 6 microorganisms-13-00345-f006:**
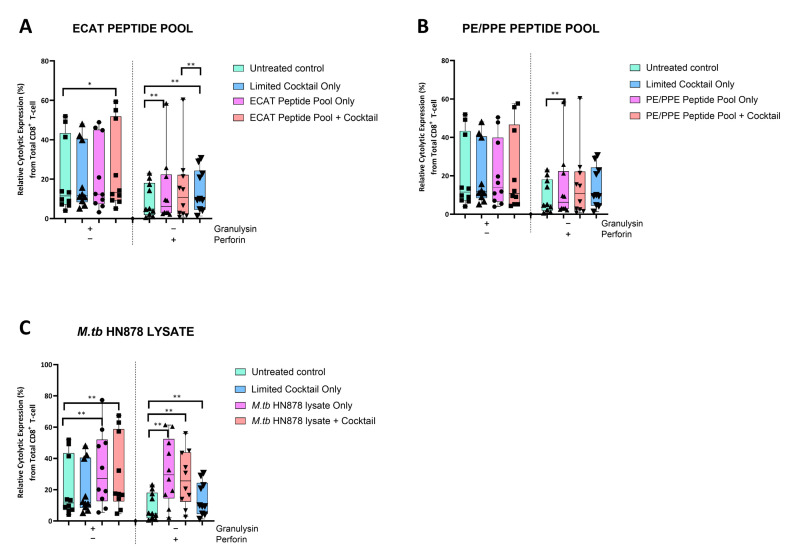
Matured mo-DCs from pre-XDR/XDR-TB patients primed CD8^+^ T-cells that expressed high levels of cytolytic markers. PBMCs were co-cultured with XDR-TB patient-derived, matured mo-DCs (*n* = 9) for seven days. The expression levels of granulysin and perforin from the DC-primed CD8^+^ T-cells, matured with either the ECAT peptide pool (**A**), the PE/PPE peptide pool (**B**) or the *M. tb* HN878 lysate (**C**) was determined by flow cytometry. The flow cytometry data were analysed using FlowJo. Statistical analysis was performed using GraphPad Prism. One-way ANOVA with Dunnett’s post-test was used to compare experimental groups to the control group (immature mo-DCs). For paired comparisons between mo-DCs matured with *M. tb* antigens only and mo-DCs with *M. tb* antigens + C, the Wilcoxon signed-rank test was applied. Significance levels are indicated as * and ** for *p* < 0.05 and *p* < 0.01 respectively. Outliers are represented by symbols, which indicate data points that fall significantly outside the interquartile range. Abbreviations: ECAT = peptide pool consisting of ESAT-6, CFP-10, AG85b, and TB10.4 peptides; PE/PPE = peptide pool consisting of PE and PPE peptides; HN878 = sonicated lysate of *M. tb* HN878; LC = limited maturation cocktail containing IFN-γ, IFN-α, IL1-β, and CD40L; C = full maturation cocktail containing IFN-γ, IFN-α, IL1-β, CD40L, Ampligen (TLR3 agonist), and R848 (TLR7/8 agonist).

**Figure 7 microorganisms-13-00345-f007:**
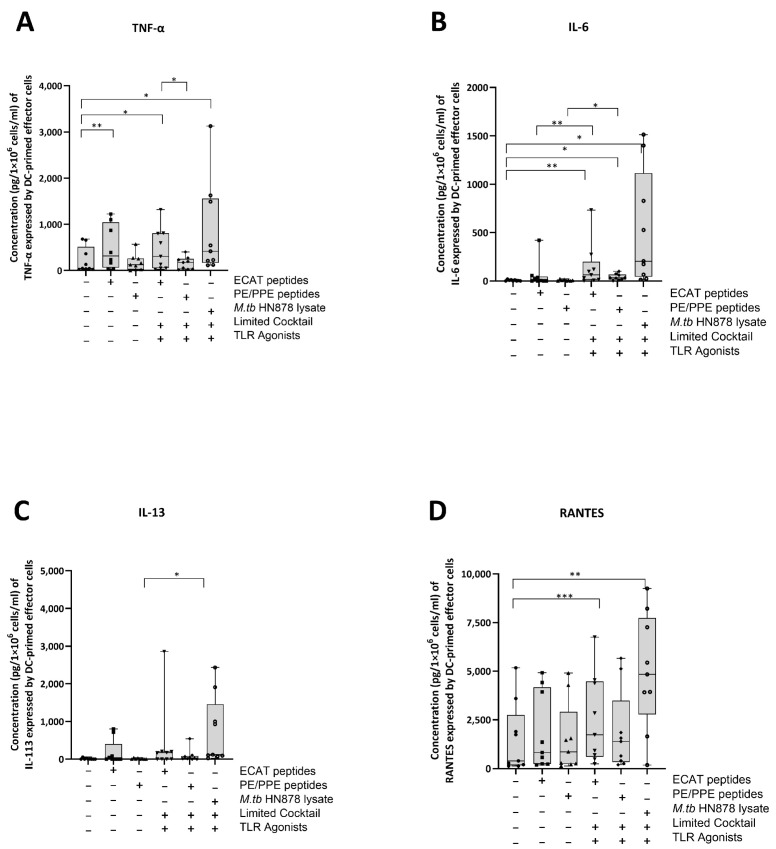
Matured mo-DCs from pre-XDR/XDR-TB patients primed effector cells that secreted high levels of soluble cytokines. PBMCs were co-cultured with XDR-TB patient-derived, matured mo-DCs (*n* = 9) for seven days. The experimental conditions are immature DC, negative control (DCs matured without *M. tb* antigens or cocktail), mo-DCs matured with ECAT only, mo-DCs matured with PE/PPE only, mo-DCs matured with HN878 only, mo-DCs matured with limited cocktail only, mo-DCs matured with ECAT + C, mo-DCs matured with PE/PPE + C, and mo-DCs matured with HN878 + C. The cell culture supernatants were analysed for the expression of the TNF-α (**A**)**,** IL-6 (**B**), IL-13 (**C**), and RANTES (**D**) using a Miliplex Luminex assay (Merck, Germany). Statistical analysis was performed using GraphPad Prism. One-way ANOVA with Dunnett’s post-test was used to compare experimental groups to the control group (immature mo-DCs). For paired comparisons between mo-DCs matured with *M. tb* antigens only and mo-DCs with *M. tb* antigens + C, the Wilcoxon signed-rank test was applied. Significance levels are indicated as *, **, and *** for *p* < 0.05, *p* < 0.01 and *p* < 0.005 respectively. Outliers are represented by symbols, which indicate data points that fall significantly outside the interquartile range. Abbreviations: ECAT = peptide pool consisting of ESAT-6, CFP-10, AG85b, and TB10.4 peptides; PE/PPE = peptide pool consisting of PE and PPE peptides; HN878 = sonicated lysate of *M. tb* HN878; LC = limited maturation cocktail containing IFN-γ, IFN-α, IL1-β, and CD40L; C = full maturation cocktail containing IFN-γ, IFN-α, IL1-β, CD40L, Ampligen (TLR3 agonist), and R848 (TLR7/8 agonist).

**Figure 8 microorganisms-13-00345-f008:**
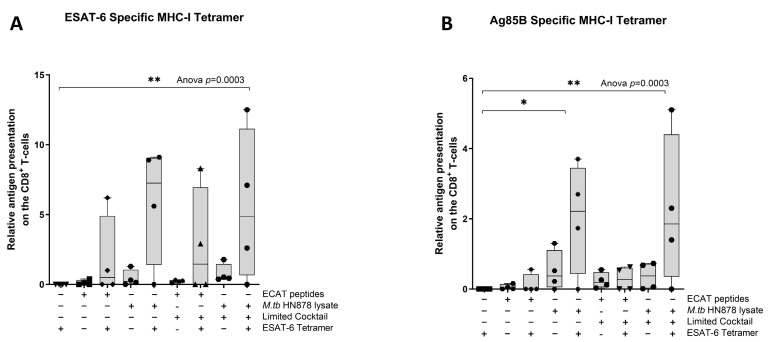
The TCRs of CD8^+^ T-cells primed with matured mo-DCs can recognize ESAT-6 and Ag85B tetramers. Autologous PBMCs were co-cultured with pre-XDR/XDR-TB patient-derived, matured mo-DCs (*n* = 4) for seven days to generate effector cells. The experimental conditions are immature DC, negative control (DCs matured without *M. tb* antigens or cocktail), mo-DCs matured with ECAT only, mo-DCs matured with HN878 only, mo-DCs matured with ECAT + C, and mo-DCs matured with HN878 + C. The CD8^+^ T-cells were stained for ESAT-6- (**A**) and Ag85B (**B**)-specific tetramers on day seven. Flow cytometry data were analysed using FlowJo. Statistical analysis was performed using GraphPad Prism. One-way ANOVA with Dunnett’s post-test was used to compare experimental groups to the control group (immature mo-DCs). For paired comparisons between mo-DCs matured with *M. tb* antigens only and mo-DCs with *M. tb* antigens + C, the Wilcoxon signed-rank test was applied. Significance levels are indicated as * and ** for *p* < 0.05 and *p* < 0.01, respectively. Outliers are represented by symbols, which indicate data points that fall significantly outside the interquartile range. Abbreviations: ECAT = peptide pool consisting of ESAT-6, CFP-10, AG85b, and TB10.4 peptides; PE/PPE = peptide pool consisting of PE and PPE peptides; HN878 = sonicated lysate of *M. tb* HN878; LC = limited maturation cocktail containing IFN-γ, IFN-α, IL1-β, and CD40L; C = full maturation cocktail containing IFN-γ, IFN-α, IL1-β, CD40L, Ampligen (TLR3 agonist), and R848 (TLR7/8 agonist).

**Figure 9 microorganisms-13-00345-f009:**
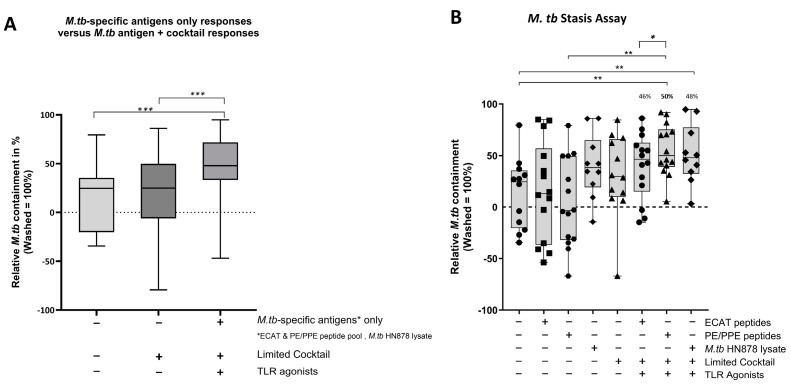
In pre-XDR/XDR-TB patients, the effector cells primed by matured mo-DCs were bactericidal to *M. tb* in vitro. *M. tb*-infected MDMs were incubated with/without DC-primed effector cells for 24 h (*n* = 14). Colony-forming units/mL were determined on Middlebrook H9/OADC agar. Mycobacterial containment is shown as a percentage and was determined relative to reference control. The dotted line represents a relative level of containment by reference control (*M. tb*-infected MDMs only, i.e., no *M. tb* containment). The grouped containment response from DC matured with *M. tb* antigen is shown in (**A**) and the containment response for all experimental conditions is shown in (**B**). The experimental conditions are immature DC, negative control (DCs matured without *M. tb* antigens or cocktail), mo-DCs matured with ECAT only, mo-DCs matured with PE/PPE only, mo-DCs matured with HN878 only, mo-DCs matured with limited cocktail only, mo-DCs matured with ECAT + C, mo-DCs matured with PE/PPE + C, and mo-DCs matured with HN878 + C. Statistical analysis was performed using GraphPad Prism. One-way ANOVA with Dunnett’s post-test was used to compare experimental groups to the control group (immature mo-DCs). For paired comparisons between mo-DCs matured with *M. tb* antigens only and mo-DCs with *M. tb* antigens + C, the Wilcoxon signed-rank test was applied. Significance levels are indicated as *, **, and *** for *p* < 0.05, *p* < 0.01, and *p* < 0.005 respectively. Outliers are represented by symbols, which indicate data points that fall significantly outside the interquartile range. Abbreviations: ECAT = peptide pool consisting of ESAT-6, CFP-10, AG85b, and TB10.4 peptides; PE/PPE = peptide pool consisting of PE and PPE peptides; HN878 = sonicated lysate of *M. tb* HN878; LC = limited maturation cocktail containing IFN-γ, IFN-α, IL1-β, and CD40L; C = full maturation cocktail containing IFN-γ, IFN-α, IL1-β, CD40L, Ampligen (TLR3 agonist), and R848 (TLR7/8 agonist).

**Table 1 microorganisms-13-00345-t001:** Demographic characteristics, disease characteristics, and haematological descriptors of enrolled pre-XDR/XDR-TB patients and LTBI participants.

	Pre-XDR/XDR-TB Patients (*n* = 25)	LTBI Participants (*n* = 18)
Median age in years (range)	34 (22–58)	46 (24–57)
Gender in %, (*n*)	Male 80% (20/25)	Male 16% (3/18)
	Female 20% (5/25)	Female 84% (15/18)
Race in %, (*n*)	Black 44% (11/25)	Black 33% (6/18)
	Mixed race 32% (8/25)	Mixed race 67% (12/18)
	Not disclosed 24% (6/25)	
Diagnosis in %, (*n*)	Pre-XDR-TB 16% (4/25)	N/A
	XDR-TB 84% (21/25)	
HIV status in %, (*n*)	Negative 68% (17/25)	Negative 100% (18/18)
	Positive 32% (8/25)	
Previous TB %, (*n*)	None 16% (4/25)	N/A
	DS-TB 12% (3/25)	
	MDR-TB 20% (5/25)	
	XDR-TB 4% (1/25)	
	Not disclosed 48% (12/25)	
CD3 median (cells/uL) (range)	HIV^−ve^: 1379 (898–2470)HIV^+ve^: 888 (411–1494)	N/A
CD4 median (cells/uL), (range)	HIV^−ve^: 835 (432–1423)HIV^+ve^: 478 (200–815)	N/A
CD8 median (cells/uL), (range)	HIV^−ve^: 445 (280–1325) HIV^+ve^: 673 (231–995)	N/A

N/A = Not applicable.

## Data Availability

All data are available for anyone with interest by contacting the University of Cape Town, Centre for Lung Infection and Immunity, through the corresponding author.

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
