# Peer review of "Autologous Human Dendritic Cells from XDR-TB Patients Polarize a Th1 Response Which Is Bactericidal to Mycobacterium tuberculosis"

_microorganisms, 2025, doi:10.3390/microorganisms13020345_

Round 1

Reviewer 1 Report

Comments and Suggestions for Authors

The study is very interesting and the results obtained are promising for finding alternatives to the treatment of patients with pre-XDR/XDR tuberculosis.

But, it is difficult to understand the study procedure. Figure 1, although more explanatory than what is described in the material and methods, does not clearly explain the entire process that has been used with the monocytes extracted from the patients (pre-XDR/XDRTb patients and participants with latent tuberculosis). This figure should be a reflection of the entire process, breaking it down and not just indicating with or without.

Author Response

Kindly refer to the attachment.

Reviewer 2 Report

Comments and Suggestions for Authors

The manuscript entitled 'Autologous human dendritic cells from XDR-TB patients polarize a Th1 response which is bactericidal to M.tb' describes efforts to develop a dendritic cell vaccine for the treatment of TB patients facing drug resistance issues. The manuscript is generally well-written, but the following points need to be addressed before publication:

1. Page 1. Section Keywords (Lines 37-38): Remove the numbers next to each keyword.

2. Page 2. Section 1 (Lines 79-80): provide a reference for the cocktail components mentioned.

3. Page 3. Section 2.4 (Lines 119-121): Consider moving the first sentence of this paragraph to the introduction section. This will provide more space to rationalize the use of mo-DC in the study.

4. Figures 2-8: The x-axis labels in the data charts are complicated. For instance, Figure 2 includes eight experimental conditions in addition to iDCs, but their descriptions are not straightforward. It is suggested to label these conditions numerically (e.g., 1–8) and include a legend as an additional panel in the figure or in the corresponding main text. Alternatively, the author may address this issue in another way.

5. Figures 2-3: The text mentions, "by one-way ANOVA with Dunnett’s post-test (immature mo-DCs was the control group) or Wilcoxon", but it does not specify which statistical significance is derived from ANOVA and which from Wilcoxon. 

6. Figures 2-6, and 8: The patient numbers are inconsistently reported:

Figure 3 and Figure 8: Patient numbers are not mentioned in the figure footnotes.

Figures 2-4, and 6: Within the same dataset, patient numbers are inconsistent. For example, in Figure 2, Panel A-D, the M.tb HN878 lysate-only condition (5th from the left) and the HN878+C condition (last) show 11 data points, whereas all other conditions show 15. Similar inconsistencies are observed in Figure 3C, Figure 4, and Figure 6B-C.

In Figure 5, the footnote reports 8 patients, but 10 data points are plotted.

Figure 8, patient number is missing in the footnote.

7. The author needs to comment on the variation in sample numbers for the experiments described in Figures 2–8.

Author Response

Kindly refer to the attachment.

Round 2

Reviewer 1 Report

Comments and Suggestions for Authors

All the modifications made by the authors have contributed to improving the presentation of the work carried out so that it can be published.

Reviewer 2 Report

Comments and Suggestions for Authors

In the revised manuscript, the authors have addressed all the points raised by the reviewers. The paper is now ready for publication. This review only requests that the authors work with the editorial office to ensure that the 'track changes' marks in the current PDF file are removed in the galley proof.